# A CSI-Based Multi-Environment Human Activity Recognition Framework

**Baha A. Alsaify** [1,*], **Mahmoud M. Almazari** [1], **Rami Alazrai** [2], **Sahel Alouneh** [2,3] **and Mohammad I. Daoud** [2]

1. Network Engineering and Security Department, Jordan University of Science and Technology, Irbid 22110, Jordan; mmalmazari16@cit.just.edu.jo
2. Computer Engineering Department, German Jordanian University, Amman 11180, Jordan; rami.azrai@gju.edu.jo (R.A.); mohammad.aldaoud@gju.edu.jo (M.I.D.)
3. College of Engineering, Al Ain University, Abu Dhabi Campus, Abu Dhabi 112612, United Arab Emirates; sahel.alouneh@aau.ac.ae
* Correspondence: baalsaify@just.edu.jo; Tel.: +962-27201000 (ext. 22503)

**Abstract:** Passive human activity recognition (HAR) systems, in which no sensors are attached to the subject, provide great potentials compared to conventional systems. One of the recently used techniques showing tremendous potential is channel state information (CSI)-based HAR systems. In this work, we present a multi-environment human activity recognition system based on observing the changes in the CSI values of the exchanged wireless packets carried by OFDM subcarriers. In essence, we introduce a five-stage CSI-based human activity recognition approach. First, the acquired CSI values associated with each recorded activity instance are processed to remove the existing noise from the recorded data. A novel segmentation algorithm is then presented to identify and extract the portion of the signal that contains the activity. Next, the extracted activity segment is processed using the procedure proposed in the first stage. After that, the relevant features are extracted, and the important features are selected. Finally, the selected features are used to train a support vector machine (SVM) classifier to identify the different performed activities. To validate the performance of the proposed approach, we collected data in two different environments. In each of the environments, several activities were performed by multiple subjects. The performed experiments showed that our proposed approach achieved an average activity recognition accuracy of 91.27%.

**Keywords:** channel state information (CSI); human activity recognition (HAR); multi-environment; support vector machine (SVM)





## 1. Introduction

Recently, the coverage of wireless signals, such as Wi-Fi signals, has expanded to cover almost every place in which people live and work. Several domains have exploited the ubiquitous wireless signals in the surrounding environment to develop real-world systems, such as Wi-Fi-based human localization and activity recognition systems [1] as well as Wi-Fi-based hand gesture recognition systems [2].

One of the leading research topics that is currently steered toward utilizing the ubiquitous Wi-Fi signals is recognizing human activities without the need to attach sensors to the subject's body. The use of the overflowing Wi-Fi signals in recognizing human activities is gaining much momentum since it does not rely on attached sensors such as accelerometers [3], gyroscope [4], or smartphones [5]. Thus, these systems do not affect in any way the movements of the subject.

Moreover, using Wi-Fi signals in recognizing human activities increases the potential of determining the activity the subject is currently performing even if no direct observation line is present between the subject and the sensing device. This might be attributed to the fact that Wi-Fi-based human activity recognition systems can operate in nonline-of-sight

(NLOS) configuration, which is considered an advantage over vision-based human activity recognition approaches that are restricted to line-of-sight (LOS) configuration [6,7]. The Wi-Fi-based human activity recognition approach also has the advantage over wearable human activity recognition approaches in the sense that it does not affect the motion of the subject.

Literature reveals that researchers have utilized two Wi-Fi-related quantities to develop human activity recognition systems, namely the Received Signal Strength (RSS) and the Channel State Information (CSI). The RSS is widely used in determining the performed activity, based on observing the changes in the power of the received signal. A main drawback of the RSS is related to the fact that it measures the power of the transmitted signal, which decays as the distance between the subject and the receiver increases. Thus, the further the distance between the subject and the measuring node, the lower the system's accuracy [8]. The other measurement used to recognize the different activities performed by a human is the CSI values. The CSI values provide a measure of the channel properties, where these properties are strongly affected by the surrounding environment and the changes that occur within the environment, such as the movements of the subjects whether these movements are as small as chest breathing movements [9] or as large as a subject walking from one point to another point within the environment [10].

In this work, we propose a CSI-based approach for recognizing several daily-life human activities. The proposed approach consists of five phases. The first phase applies a denoising method and a dimensionality reduction procedure to the recorded CSI signals. This phase is needed to reduce the interference encapsulated within the recorded CSI signals and, at the same time, reduce the amount of data without losing the information encapsulated within the available data. In the second phase, a segmentation algorithm is applied to the CSI signals obtained from the first phase to extract the segments of the signals that represent intervals during which the subjects performed various activities. After that, in the third phase, the segments obtained from the second phase are cropped from the raw CSI signals, and the noise reduction procedure, which is applied in the first phase, is applied to the extracted activity segments. In the fourth phase, a set of features is extracted from the activity segments. In addition, a feature selection procedure is applied to select the best combination of features that can be used to recognize different human activities. Finally, the fifth phase utilizes the extracted features to construct support vector machine (SVM) classifiers to distinguish between the CSI signals associated with different human activities. To validate the performance of our proposed approach, we have collected a large dataset composed of CSI signals acquired from 20 subjects while performing six different daily-life human activities, including walking, falling, sitting on a chair, picking up a pen from the ground, and turning. The dataset was collected in two different environments to evaluate the capability of our proposed approach to recognize human activities that are performed in different environments. In addition, in this work, we are performing a leave-one-subject-out cross-validation technique to test the ability of the developed models to recognize activities from new subjects.

The main contribution of this work can be summarized in the following points:

- We addressed a multi-environment human activity identification problem.
- We evaluated the proposed system using a leave-one-subject-out manner in which no activity trace from that subject is used to train the classification model.
- A novel segmentation method to remove the part of the signal that contains no-movements is developed.
- A methodology for denoising the collected CSI values, which are obtained from the Wi-Fi exchanged packets, is introduced.
- A large number of handcrafted features were investigated to determine their effectiveness in recognizing the different performed human activities in a multi-environment domain.
- To add to the credibility of this work, ample experiments are conducted using the publicly available dataset to demonstrate the effectiveness of the proposed methodology in determining the differently performed human activities.

The rest of this paper is organized as follows: In the "Related Work" Section, we provide a summary of the recent work proposed in the area of activity recognition. After that, in the "Background" Section, we provide a brief background about the CSI. In the "Methodology" Section, we provide a detailed description of the acquired dataset including the performed activities and the environments in which the datasets were recorded. A detailed description of the extracted features and the process we used to collect these features are also provided in the "Methodology" Section. In the "Results" Section, we present and discuss the results obtained using our proposed approach. Finally, the conclusions are provided in the "Conclusions" section.

### 1.1. Related Work

Literature reveals that the existing human activity recognition (HAR) systems have utilized various sensors and techniques. Particularly, these existing HAR systems can be generally divided into the following two main categories: wearable approaches and non-wearable approaches. A graphical representation of the different approaches used to recognize human activities is provided in Figure 1.

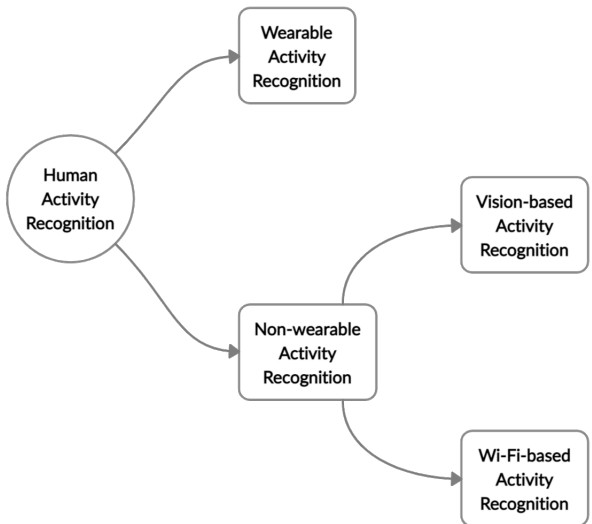

**Figure 1.** Overview of human activity recognition approaches.

### 1.1.1. Wearable Activity Recognition

We refer to the HAR systems in which the subject has to wear some hardware on their body by wearable activity recognition approaches. These approaches suffer from two main drawbacks. First, it produces a burden on the body of the subject in the form of the discomfort the subject will feel from wearing such a device. Second, the accuracy of the system is dependent on the location of the sensor on the body of the subject. Much work has been done in this area, such as the work presented in [11], which attempted to determine the activity being performed by studying the quality of the harvested Kinetic energy. The approach presented in [11] was evaluated using two datasets. The first data set is a public dataset presented in [12], while the second dataset is their own generated KEH dataset. The evaluation of the system showed that the placement of the sensor on the body of the subject is vital to achieve high activity recognition accuracy, which was taken into consideration when recording data for their own generated dataset.

Another work that focuses on the data acquired from an accelerometer sensor is presented in [13]. In this study, a dataset [14] with data from four subjects was collected and analyzed. Despite the high recognition accuracy that was reported in [13], the small size of the utilized dataset imposes the requirement to conduct further analyses to validate the performance of the proposed approach. Similarly, the approaches presented in [15,16] attempted to recognize human activity by analyzing the accelerometer data collected using

smartphones or specially built hardware. A more comprehensive survey on the existing studies that focused on HAR using accelerometer data can be found in [17].

1.1.2. Non-Wearable Activity Recognition Approaches

We refer to the approaches by which the activities of a human can be recognized without placing any sensor on the subject as *non-wearable activity recognition approaches*. Non-wearable approaches can be generally grouped into two main categories: vision-based approaches and Wi-Fi-based approaches [18,19].

Vision-based activity recognition was the first approach by which researchers attempted to recognize human activities. The work of [20] utilized low-cost depth cameras to recognize human activities. The effectiveness of their approach was tested on three public benchmark datasets, including: MSR daily activity 3D [21], MSR action pair dataset [22], and MSR action 3D dataset [23]. The approach presented in [20] partitioned the body into moving parts, with each of these parts used to generate a set of features. Ref. [24] proposed a compact representation of the human postures via histograms of the 3D location of joints. Hidden Markov models were used to analyze the extracted feature vectors to build the classification models. A novel skeleton representation was proposed by [7]. Inspired by the observation that the geometric relationship between the body parts holds much more meaning than their abstract location, a skeletal representation was developed. In [6], using the chain coding technique, the silhouettes from different frames are bounded together and converted into joint points to provide an online human action recognition system.

Despite the promising results attained by vision-based approaches, there are several limitations that reduce the potentials of using these approaches in real-world applications, including: (1) The need to record the activities of the subject implies that there is an invasion of the privacy of the subject. (2) These approaches fail to recognize the subject's activity if the subject was located behind an obstacle. As a remedy to the aforementioned limitations, researchers have investigated other techniques, such as Wi-Fi signals.

Wi-Fi-based activity recognition came to light due to the increasing popularity of the Internet of Things (IoT) concept. Almost all devices in one's household can now connect with each other and with the Internet via wireless connectivity. Researchers noticed this trend and utilized these wireless signals to recognize the activities the subject performs. Particularly, researchers have utilized the CSI values embedded within the exchanged wireless packets to recognize human activities. Unlike the vision-based approaches, wireless signals can penetrate objects; thus, the subject's activity can be recognized even with the existence of obstacles. Furthermore, HAR systems that relies on wireless signals can preserve the privacy of subjects. In this regard, researchers have utilized the CSI signals to recognize human emotions [25], sleep monitoring [26], vital signs monitoring [27], human–human interactions recognition [28,29], and human identification [30,31].

An example on CSI-based HAR systems is the approach presented by [32], in which they developed an activity recognition system composed of two main models. Specifically, the first model determines the speed at which the subject is performing the activity, while the second model maps the speed from the first model with the activity being performed. The researchers in [32] attempted to determine the activity from the speed of the subject, which was determined using the CSI information contained in the Wi-Fi signals. Similarly, Ref. [33] attempted to recognize human activities in an indoor environment. The first problem they tried to solve was determining if an activity has occurred or not, which was solved using a two-level decision tree. The first level detects activities, while the second level utilizes a deep neural network to determine the activity detected in the first level. An accuracy between 96% and 98% was achieved by increasing the number of receiver antennas. In another work by [34], an HAR approach was proposed based on learning representative features in two directions, namely forward feature selection and backward feature selection, and then assign weights to these features based on their importance in activity recognition. To test their hypothesis, the researchers used the dataset presented in [35]. One of the recent publications that tackled the problem of CSI-based HAR problem

can be found in [36]. In this work, a multi-environment CSI-based HAR approach was devised compared to other work that can be found in the literature which focuses on the developing single-environment HAR system.

*1.2. Background*

Currently, wireless technology is moving forward in adopting the concept of Multiple Input Multiple Output (MIMO) in communication systems that comprises multiple transmit–receive pairs of antennas. Each transmit–receive pair of antennas creates a communication channel that utilizes a modulation technique to transmit coherent information over the established channel. Among the different modulation techniques, the Orthogonal Frequency Division Multiplexing (OFDM) [37] has been widely used by several Wi-Fi standards such as 802.11a, 802.11n, and 802.11ac to encode and transmit information from a transmitter to a receiver. OFDM creates several closely spaced communication channels to spread the data being transmitted over it. OFDM can be viewed as a digital-subcarrier modulation technique. Each of these subcarriers can be characterized by using the CSI.

Assuming the existence of *m* transmitters and *n* receivers, each transmit–receive pair will create a communication channel that can be characterized by 30 CSI components as follows:

$$H_{t,r} = [H_{t,r}^1, H_{t,r}^2, \ldots, H_{t,r}^{30}], \tag{1}$$

where $t \in \{1, \ldots, m\}$ and $r \in \{1, \ldots, n\}$. Each of the 30 CSI components is defined by two main attributes, namely the amplitude and phase, as follows:

$$H_{t,r}^i = A_{t,r}^i e^{j\theta_{t,r}^i}, \ i \in [1, \ldots, 30], \tag{2}$$

where $A$ is the amplitude, $\theta$ is the phase, $t$ is the transmitter, $r$ is the receiver, and $i$ is the CSI subcarrier index of an OFDM channel between a pair of transmitter and a receiver.

The CSI describes the link state information carried in each transmitted wireless packet. These packets are affected by the environment conditions as follows:

$$R_x = H \times T_x + N, \tag{3}$$

where $R_x$ is the received signal vector where each entry of this vector represents a packet, $T_x$ is the transmitted signal vector where each entry of this vector represents a packet, $H$ is the channel state information matrix, and $N$ is the additive noise. In this study, we refer to the CSI values recorded at a particular time index for all subcarrier frequencies and all pairs of transmit–receive antennas as packet.

Assuming that there are *m* transmitters and *n* receivers in the MIMO system, the CSI components ($H$) of any of the received packets can be expressed as follows:

$$H = \begin{bmatrix} H_{1,1} & H_{1,2} & \ldots & H_{1,n} \\ H_{2,1} & H_{2,2} & \ldots & H_{2,n} \\ . & . & \ldots & . \\ . & . & H_{t,r} & . \\ . & . & \ldots & . \\ H_{m,1} & H_{m,2} & \ldots & H_{m,n} \end{bmatrix} \tag{4}$$

A technical diagram illustrating the CSI values exchanged between a transmit–receive pair of devices through time is provided in Figure 2. More information about CSI can be found in [38,39].

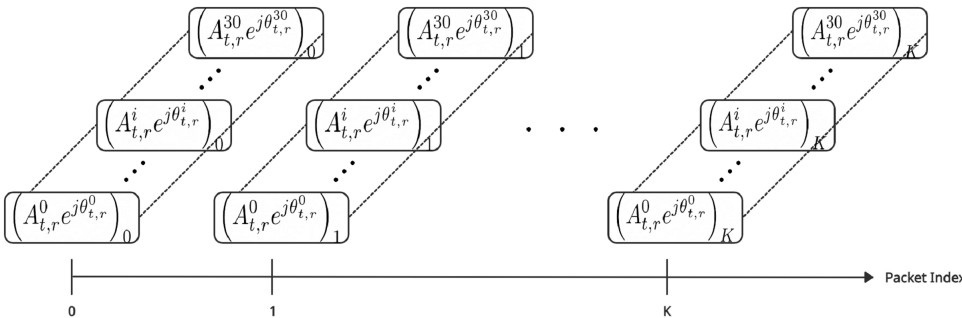

**Figure 2.** Technical diagram of received packets from a transmitter–receiver pair.

## 2. Materials and Methods

In this section, a description of the proposed system is provided. In addition, the data collection process, a description of the data collection environments, and a description of the different performed activities are provided. A graphical representation of the system and its different stages are provided in Figure 3.

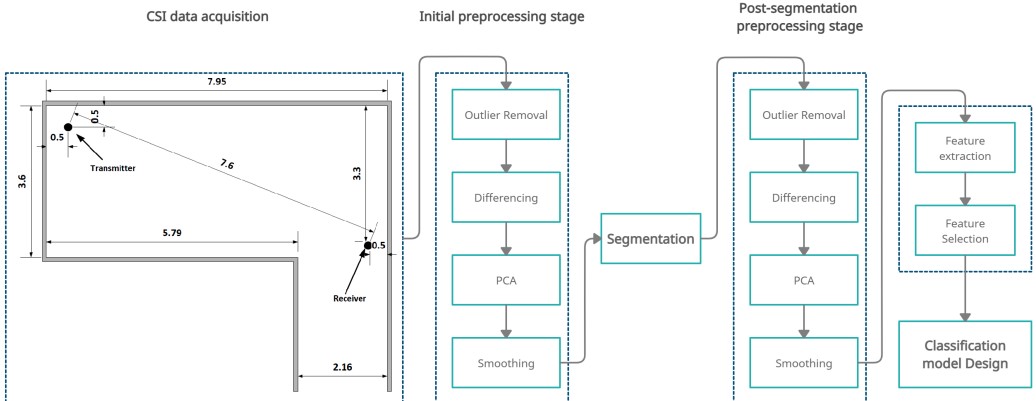

**Figure 3.** The structure of the proposed HAR system.

### 2.1. Experimental Setup

In this work, we have designed several experiments to validate the performance of our proposed CSI-based HAR approach. Particularly, in each experiment, the subjects were asked to perform a set of pre-explained tasks in the area between a transmit–receive pair of antennas. During the experiments, a beep sound was used to guide the subjects and inform them on when to start a specific task. The transmitter and the receiver were both retrofitted with "Intel Ultimate N Wi-Fi Link 5300" to extract the CSI values from the transmitted packets using the CSI tool described in [40]. The transmitter and the receiver were configured according to the settings provided in Table 1. As shown in Table 1, we have a one-to-many architecture in which only one antenna is transmitting the packets, and three antennas are receiving the packets.

### 2.2. Environment Description

The performed experiments were conducted in the two environments: an office and a university hallway. In the first environment, the subjects performed the experiments inside an office, while, in the second environment, students and university employees were moving in the vicinity of the hallway along with the subject performing the experiments. The differences between the two environments can be summarized as follows: First, the distance between the transmitter and the receiver is different between the two environments. Having a change in the distance will allow us to study its effect on the overall system accuracy. Second, the environment configuration is different. The office environment has a square configuration, while the hallway environment has an L-shaped configuration.

**Table 1.** Transmitter and receiver settings.

| Setting | Value |
|---:|---|
| Frequency | 2.4 GHz |
| Channel Number | 3 |
| Transmission Mode | Injection |
| Channel Width | 20 MHz |
| Sampling Rate | 320 Packets/second |
| Packet Size | 1 Byte |
| Modulation Type | MCS0 |
| Data Rate | 6 Mbps |
| Encoding Technique | BPSK |
| Coding Rate | 1/2 |
| Spatial Streams | SIMO |
| Architecture | 1 transmitter and 3 Receivers |

### 2.2.1. The Office Environment

In this environment, the experiments were performed in an office where only authorized people could enter. A sketch and a photo of the room in which we performed the experiments are provided in Figure 4. All the dimensions shown on the sketch are measured in meters. With the exception of the walking activity, to perform an activity, the volunteering subject remained in the middle between the transmitter and the receiver. Each subject was provided with general instructions on how to perform the experiment. For example, we instructed the subject to pick up a pen from the ground within the allowed time. However, we did not specify how the subject should pick the pen up (the subject may bend down and pick the pen up, the subject may squat to pick it up, or the subject can pick it up in any other way he/she sees fit). In this environment, the distance between the transmitter and the receivers was fixed at 3.7 m. No obstacle between the subject and the transmitter or the receiver was present, so all the experiments were performed in a line-of-sight (LOS) configuration.

### 2.2.2. The Hallway Environment

In this environment, the experiments were performed in a university hallway where students constantly moved around. A sketch and a photo of the hallway are shown in Figure 5. Similar to the office experiments, the subjects were instructed to perform the required tasks in the middle between the transmitter and the receiver. No obstacles were present, so all the experiments were performed in an LOS architecture.

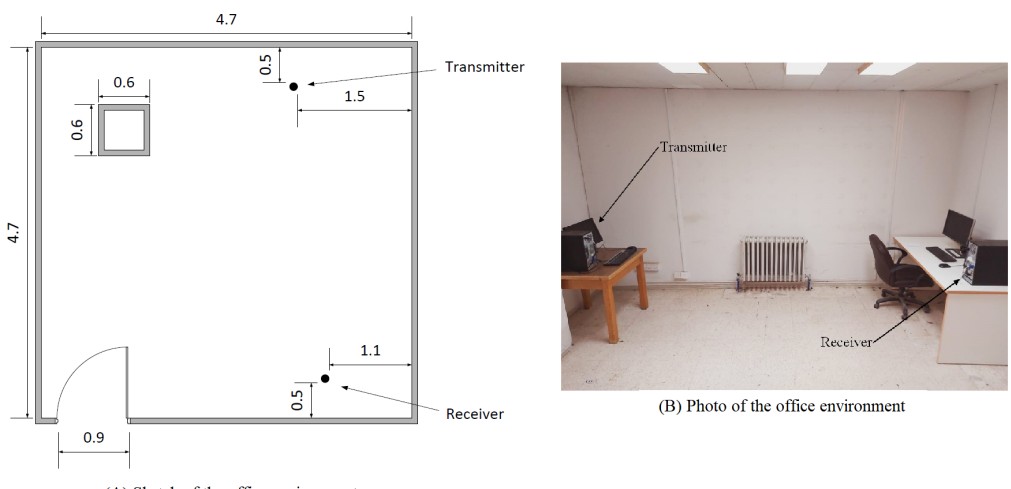

(A) Sketch of the office environment

(B) Photo of the office environment

**Figure 4.** The office environment used for data collection. All dimensions are in meters.

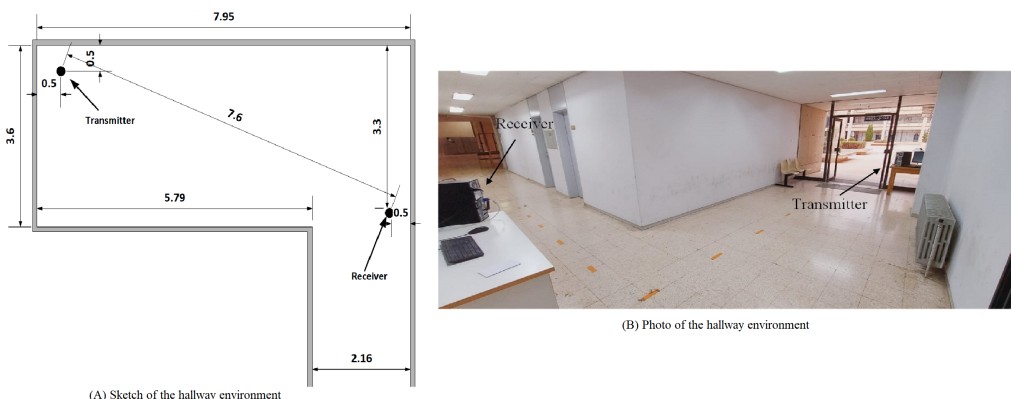

(A) Sketch of the hallway environment

(B) Photo of the hallway environment

**Figure 5.** The hallway environment used for data collection. All dimensions are in meters.

## 2.3. Activity Description and Data Collection

For each of the two environments, ten subjects volunteered to perform the experiments. Each subject was asked to perform five experiments, where each experiment was repeated 20 times. The total number of performed trials in each environment is equal to 1000 trials, and since three antennas receive the signals, the total number of observations is equal to 3000 observed trials. The IRB office at the King Abdulla University Hospital (KAUH) and the Jordan University of Science and Technology approved this research (19/110/2017). A written consent indicating that the personal information of the subjects will not be disclosed and that they have the right to stop participating in any of the experiment if they chose to was acquired from each of the subjects.

Once all the experiments were performed, eleven activities were identified according to the time stamp associated with the beep sound used to guide the subjects during the experiments. Figure 6 shows the timing diagrams of the activities identified within the performed experiments. Moreover, Table 2 provides a summary of the activities identified within the performed experiments.

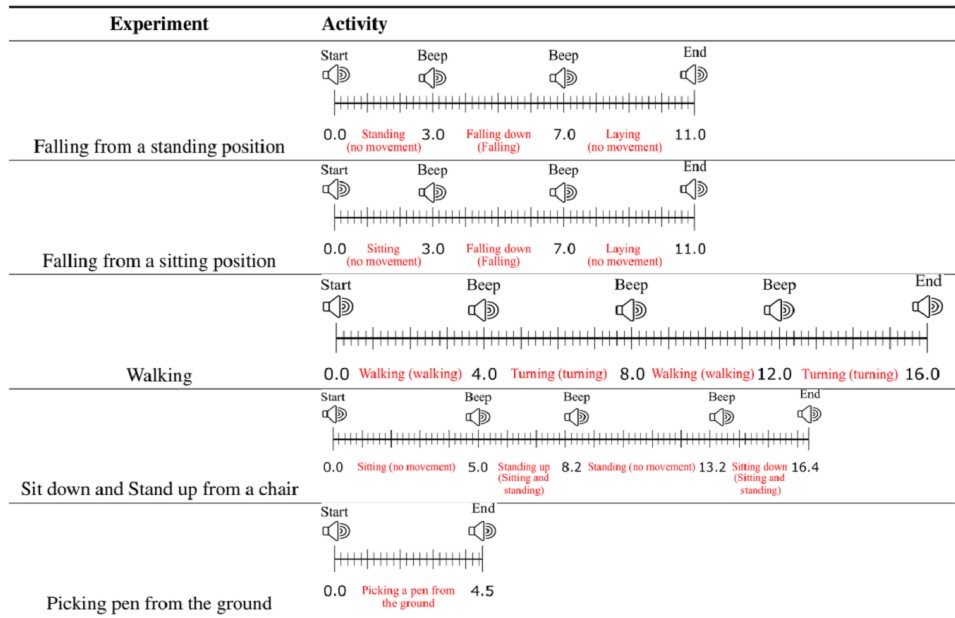

**Figure 6.** The timing diagrams of the identified activities.

**Table 2.** Summary of the identified activities within the performed experiments.

| Activity Label | Activity Name | Description |
|:---:|:---:|:---:|
| A1 | No movement | This activity comprise of standing, sitting, or laying on the ground. |
| A2 | Falling | This activity comprise of falling from a standing position or falling from a chair. |
| A3 | Walking | Walking between the transmitter and the receiver. |
| A4 | Sitting down on a chair or standing up from a chair | This activity comprise of sitting on a chair or standing up from a chair. |
| A5 | Turning | This activity comprise of turning at the transmitter or at the receiver. |
| A6 | Pick up a pen from the ground | Pick up a pen from the ground. |

The data collection process was performed at different times and dates. Some subjects performed the required tasks and did not take breaks between the tasks, while other subjects performed the tasks at different times and took breaks between them. The work presented in [41] provides an in-depth look at the data we collected and used in this work.

*2.4. Data Pre-Processing*

The recorded CSI values need to be processed before it can be used due to the following reasons. First, the recorded CSI values are noisy [30]. Hence, the CSI signals need to be filtered to reduce noise before it can be processed further. Second, when recording the signals that describe a specific activity, the subjects did not start moving at the beginning of the signal recording indicator, and not all of the subjects stopped moving exactly when we stopped recording. Thus, a segmentation method to eliminate these steady-state time intervals (i.e., times at which the subject is not moving at the beginning and at the end of the recording session) is needed. Finally, using all 30 sub-carriers from each stream will increase the processing time and the complexity of the applied procedures. Principal component analysis (PCA) was used to select the effective sub-carriers and discard the sub-carriers that are less informative.

That being said, we apply a three-stage preprocessing procedure to the CSI signals before extracting the features. The three stages are: the initial preprocessing stage, the segmentation stage, and the post-segmentation preprocessing stage. Figure 7 shows a block diagram of the employed three-stage preprocessing procedure.

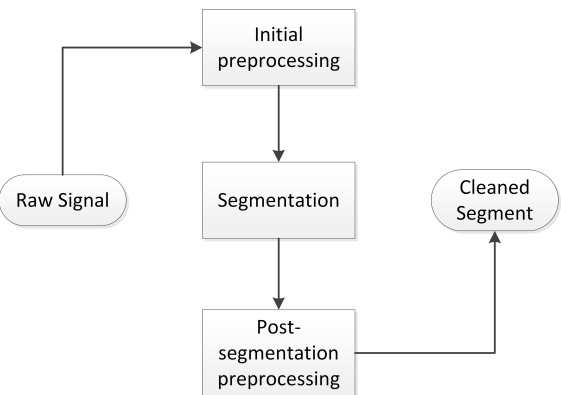

**Figure 7.** Preparing the signal for feature extraction.

### 2.4.1. Initial Preprocessing Stage

In this stage, we apply four techniques to reduce the noise and outliers in the CSI signals. These techniques are described below:

A. Outliers Removal: This technique is used to remove any value that can be considered as an outlier. Outlier values can occur for several reasons: the noise present in the environment incorrect hardware reading, and the sudden change in the environment are possible reasons for such outliers. To exclude these outliers, the Hampel filter [42] was used. The Hampel filter utilizes a sliding window that scans the CSI signals and removes any reading that is more than three times the standard deviation ($3\sigma$) by replacing that value with the mean of the values within the sliding window. An example of a processed signal using the Hampel filter is shown in Figure 8.

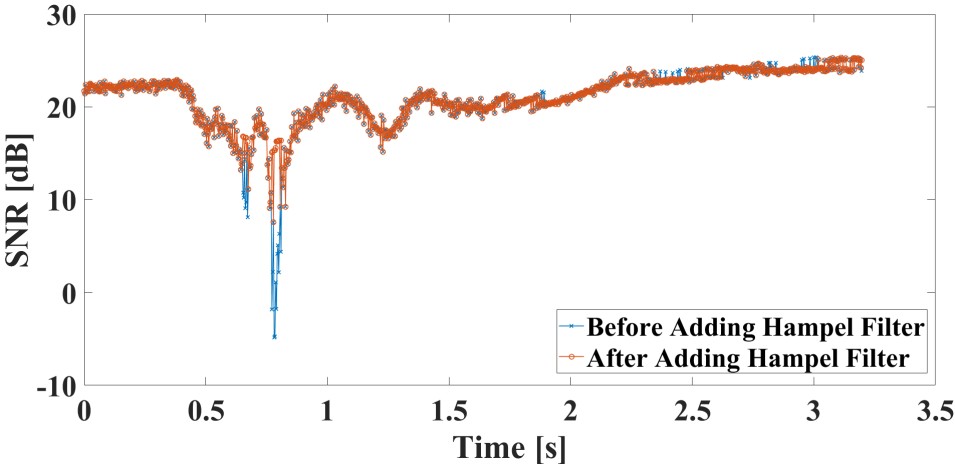

**Figure 8.** The effect of using the Hampel filter.

B. Converting to Stationary signal: A stationary signal is a term used to describe a signal with constant or slowly changing statistical properties as discussed in [43]. In this work, the signals we are dealing with are all non-stationary signals [44]. The term non-stationary signal refers to a signal with statistical values (mean, median, variance) that are changing over time. To transform the non-stationary signal into a stationary signal, the differencing procedure can be applied by subtracting each value of the time-series signal from the one before it as follows [45]:

$$y_{new} = y_t - y_{t-1} \tag{5}$$

where $y_t$ represents a reading at time $t$, $y_{t-1}$ represents the previous data reading, and $y_{new}$ represents the new reading that will replace $y_t$.

To determine if this step is necessary or not, we performed the Augmented Dickey–Fuller (ADF) test [46]. The ADF test determines if the signal at hand is stationary or not by testing the null hypothesis ($\phi = 0$), which refers to a non-stationary signal. To determine if the signal is stationary, we look for a *p*-value less than or equal to 0.05. An example of the performed ADF test is provided in the "Before Segmentation Processing" Section.

C. Data Reduction: Each receiving antenna records 30 CSI-subcarriers in each of the communication channels. The use of all 30 subcarriers will create a strain on the system. To reduce the number of signals to be analyzed and at the same time to retain the valuable information in these signals, we use the Principal Component Analysis (PCA) [47], which is considered a dimensionality reduction method. To determine the number of principal components to be selected, the following two metrics were considered: (1) The data variance in the selected principal components; (2) The errors

in the segmentation process. A detailed description of each metrics is provided in the "Before Segmentation Processing" section.

D.  Signal Smoothing: For each of the principal components we selected in the previous stage, we need to reduce the noise in that signal through a smoothing technique. One of the most known techniques to remove such noise, especially in the area of image processing, is known as Gaussian filtering [48]. The basic approach to smooth a signal is by defining a Gaussian window with the point we want to smooth in the middle of the window. To generate the smoothed point, each actual data point has a related Gaussian weight (the summation for the weights must be equal to 1). These Gaussian weights are then multiplied by the actual data values, and the summation of the multiplication results is assumed to be the new value after smoothing.

### 2.4.2. The Segmentation Stage

When performing the experiments, each subject was instructed to perform a particular activity upon hearing a beeping sound. The recording device will also start recording the Wi-Fi packets once this sound is generated. The main issue was that the subject does not start moving precisely when they hear the produced beep sound. In other words, one subject might start the activity 100 ms after hearing the sound, and another subject might start 300 ms after hearing the sound. Because of this delay, we have developed a mechanism by which the no-movement period of the signal can be separated from the period of the signal associated with actual activity. We achieved the previous requirement by using a segmentation algorithm. The segmentation algorithm takes as input a collection of signals which describes the same activity trial. For each activity, the algorithm will generate two points: the beginning and the end of an activity segment.

The segmentation process starts by scanning each principal component and selecting the points at which a change occurs on them based on the standard deviations, and root mean square values, which is based on the algorithm presented in [49,50]. In some cases, the proposed segmentation process was not able to determine the points at which a change occurs, and it will issue a segmentation error. A solution to this issue was found by using the MAX and MIN values found in the signal. Section "segmentation" shows all possible segmentation outcomes and the solution to the case at which a segmentation error had occurred.

A visual example of the signal before and after the segmentation process is provided in Figure 9. Figure 9A shows the signal before applying the segmentation process, in which the signal contains the samples associated with both the activity and steady-state periods. Figure 9B shows the resultant signal after applying the segmentation process and removing the samples associated with the steady-state period.

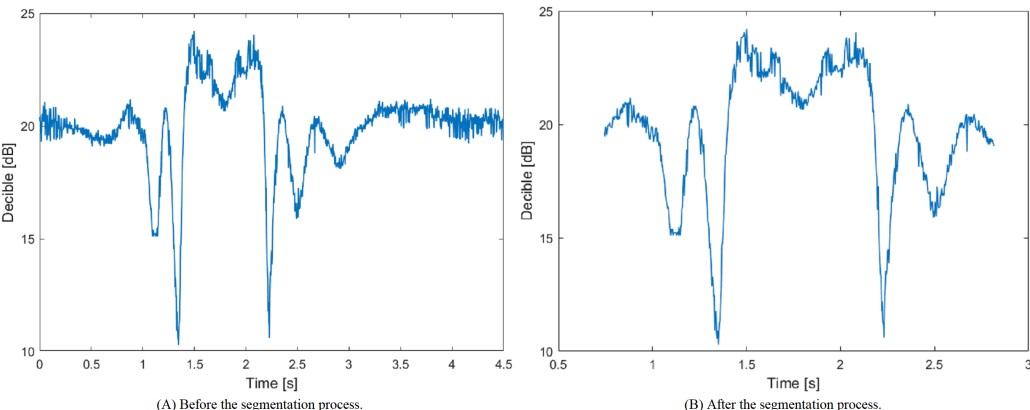

(A) Before the segmentation process.　　　(B) After the segmentation process.

**Figure 9.** The effect of the segmentation stage on a recorded signal.

### 2.4.3. The Post-Segmentation Preprocessing Stage

Similar to the initial preprocessing stage described in Section "Initial Preprocessing", this stage aims to clean and smooth the signals, but instead of manipulating the whole signal, only the signal's subset that contains the activity is processed and manipulated. The reason behind performing the preprocessing stage for a second time is to remove any influence the no-movement part of the signal has on the features to be extracted next. After determining the activity segment, the original signal can be divided into three intervals:

- The first interval represents the interval until the subject begins the activity. During this time interval, we expect the subject to remain stationary;
- the second interval represents the segment in which the activity occurred; and
- the third interval represents the interval from the end of the activity until the subject is instructed to finish the activity. Similar to the first interval, the subject is expected to remain stationary during the third interval.

Only the second interval is selected once we successfully divided the signal, while the first and third intervals are discarded. After that, we process the second interval by applying the techniques introduced in the initial preprocessing stage, described in Section "Initial Preprocessing".

### 2.4.4. Feature Extraction

In order to recognize the different activities based on the processed activity segments, we have extracted a total number of 565 features divided into 16 sets: (1) Time series features [46]. (2) The signal-to-noise ratio (SNR). (3) Frequency series features. (4) Linear predictor coefficients (LPC) [51]. (5) Line spectral frequencies (LSF) [52]. (6) Distribution tests [53]. (7) Location tests [54]. (8) Dispersion tests [55]. (9) Stationary tests set [56]. (10) Correlation [46]. (11) Causation [57]. (12) Heteroscedasticity [58]. (13) Cointegration set [59]. (14) Instantaneous frequency set [60]. (15) Spectral entropy set. (16) Audio features set [61]. A summary of the extracted features can be found in Table 3.

**Table 3.** Extracted features.

| Set of Features | Feature | Description |
|---|---|---|
| Time series features | Mean | Several mean values were collected for: signal elements, linear weighted signal, power values in the signal, linear weighted power values, difference in adjacent elements, absolute difference in adjacent elements, power of difference in adjacent elements, and the absolute difference of differences. |
| | Skewness | Signal skewness |
| | Kurtosis | Signal kurtosis |
| | MAD | Signal mean absolute deviation |
| | Crossing rate | for 2, 1, 0, −1, −2 values |
| | RSSQ | Root sum of squares |
| | MAX | Maximum element in the signal |
| | MIN | Minimum element in the signal |
| | MAX-MIN | Difference between the max and min |
| | Quartile values | The first, second (also known as median), and the third quartile |
| | IQR | Inter quartile range (Q3–Q1) |
| | Normalized count of samples | For both the samples higher and lower than the mean of the signal. |
| | Element index | index of the first occurrence of the max and min signal elements |
| | cross-cumulants | second, third, and fourth order |
| | Number of peaks | based on a prominence value of 0.1 and 0.2 |
| | ARMA model parameters | Values for the AR, MA, and the model's constant |
| SNR | SNR | Signal to noise ratio |
| Frequency series features | Median frequency | The signal's median angular frequency |
| | Mean frequency | The mean angular frequency of; the entire signal, the first 10% of the signal, the next 10% of the signal, …, the last 10% of the signal |
| | Bandwidth | The occupied bandwidth of the signal |
| | Power bandwidth | Compute the bandwidth of the part that is 3-dB from the peak value |
| Linear predictor coefficients | LPC | The second, third, and fourth coefficients |
| Line spectral frequencies | LSF | Obtained from the prediction polynomial with coefficients obtained from the LPC |

**Table 3.** *Cont.*

| Set of Features | Feature | Description |
|---|---|---|
| Distribution tests | Anderson–Darling test | The test result at a 0.05 level and the *p*-value |
| | chi-square test | The test result at a 0.05 level and the *p*-value |
| | Durbin–Watson test | The *p*-value |
| | Jarque–Bera test | The test result at a 0.05 level and the *p*-value |
| | Kolmogorov–Smirnov test | The test result at a 0.05 level and the *p*-value |
| | Lilliefors composite test | The test result at a 0.05 level and the *p*-value |
| | Runs test for randomness | The test result at a 0.05 level |
| Location tests | Wilcoxon rank sum test for equal medians | The test result at a 0.05 level and the *p*-value |
| | Wilcoxon rank sum test for zero medians | The test result at a 0.05 level and the *p*-value |
| | Sign test for zero medians | The test result at a 0.05 level and the *p*-value |
| | One-sample and paired-sample *t*-test | The test result at a 0.05 level and the *p*-value |
| | two-sample *t*-test | The test result at a 0.05 level and the *p*-value |
| Dispersion tests | Ansari–Bradley two-sample test | The test result at a 0.05 level and the *p*-value |
| | two-sample F test | The test result at a 0.05 level and the *p*-value |
| Stationary tests | Augmented Dickey–Fuller test | The test result at a 0.05 level and the *p*-value |
| | KPSS test | The test result at a 0.05 level and the *p*-value |
| | Leybourne–McCabe test | The test result at a 0.05 level and the *p*-value |
| | Phillips–Perron test | The test result at a 0.05 level and the *p*-value |
| | Variance ratio test | The test result at a 0.05 level and the *p*-value |
| | Paired integration/stationarity tests | The test result at a 0.05 level and the *p*-value for both the signal and the signal differences. |
| Correlation | Auto-correlation | Extract the time-series features from the auto-correlation sample signal |
| | Partial auto-correlation | Extract the time-series features from the partial auto-correlation sample signal |
| | Cross-correlation | Extract the time-series features from the cross-correlation between first and second half of the signal |
| | Linear-correlation | Linear-correlation between the first and second halves of the signal |
| | Ljung-Box Q-test | The test result at a 0.05 level and the *p*-value |
| | Belsley collinearity diagnostics test | Strength of collinearity in the first and second halves of the signal |
| Causation test | Granger causality tests | The test result at a 0.05 level and the *p*-value |
| Heteroscedasticity | Engle test | The test result at a 0.05 level and the *p*-value |
| Cointegration | Engle–Granger cointegration test | The test result at a 0.05 level and the *p*-value |
| | Johansen cointegration test | The test result at a 0.05 level and the *p*-value for the first and second halves of the signal |
| Instantaneous frequency | Instantaneous frequency | Extract the time-series features from the temporal derivative of the oscillation phase divided by $2\pi$ |
| Spectral entropy | Spectral entropy | Extract the time-series features from the spectral entropy of the signal |
| Audio features | Spectral Centroid | Extract the time-series features from the spectral centroid of the signal |
| | Spectral Crest | Extract the time-series features from the spectral crest of the signal |
| | Spectral Decrease | Extract the time-series features from the spectral decrease of the signal |
| | Spectral Entropy | Extract the time-series features from the spectral entropy of the signal |
| | Spectral Flatness | Extract the time-series features from the spectral flatness of the signal |
| | Spectral Flux | Extract the time-series features from the spectral flux of the signal |
| | Spectral Kurtosis | Extract the time-series features from the spectral kurtosis of the signal |
| | Spectral Rolloff | Extract the time-series features from the spectral rolloff of the signal |
| | Spectral Skewness | Extract the time-series features from the spectral skewness of the signal |
| | Spectral Slop | Extract the time-series features from the spectral slop of the signal |
| | Spectral Spread | Extract the time-series features from the spectral spread of the signal |
| | MEL Spectrum | Extract the time-series features from the MEL spectrum of the signal |
| | Estimation of the fundamental frequency | Extract the time-series features from the fundamental frequency of the signal |
| | Signal loudness | Total loudness of the signal |

### 2.4.5. Feature Selection

Once all the features have been extracted from the processed signals, we apply the minimum Redundancy Maximum Relevance (mRMR) [62] feature selection algorithm to rank the available features based on their importance in determining the activity being performed.

We start by removing the feature vectors related to the steady-state and the walking classes as we are focusing on in-place activities. Next, we apply the mRMR feature selection algorithm, which generates two arrays. The first array contains the score of the features, while the second array contains the features ordered based on the scores obtained in the first array. To select only one instance, we apply the following equation:

$$FeatureToSelect = featureID(mod\ featureCount)$$

Once we rank the features from the most essential (highest score) to the least essential (lowest score), we compare the score of the features with a threshold and select only the features with scores higher than that threshold. We used different threshold values to determine the features that result in the highest system accuracy.

### 2.4.6. Classification Model

After determining the relevant features, each observation will be characterized by a set of features instead of the CSI values. These feature vectors will be used to build the classification models responsible for determining the activity being performed. To perform the activity recognition phase, in this work, we are using the SVM classifier. The used SVM classifier has a Gaussian radial basis function (RBF) kernel with uniform prior for the different classes. We are using a one-vs.-one (OvO) approach to convert the multi-class classification problem into multiple binary classification problems. For each of the developed experiments, to optimize the SVM parameters, a coarse grid search followed by fine grid search is used. The decision behind using the SVM classifier in this work over the other classification techniques was due to the following reasons: First, the SVM classifier is the most commonly used classification technique in many classification domains, one of which is the human activity recognition domain [63–65]. Second, the SVM classifier is considered stable in the sense that it is not affected by noise compared to other classification algorithms such as decision trees. Furthermore, SVM classifier with a Gaussian RBF kernel has few hyper-parameters, namely the regularization parameter (C), and the RBF width ($\sigma$), to be optimized, which effectively reduces the fine-tuning time.

## 3. Results

In this section, the results of the experiments that were carried out to validate the performance of the proposed approach are provided.

### 3.1. Data Preparation

After collecting and recording the data packets that contain the CSI measurements, we processed the gathered signals by using the technique shown in Figure 7 and described in the "Initial Preprocessing" Section.

#### 3.1.1. Initial Preprocessing Stage

For each of the recorded signals, we removed the outliers using the Hampel filter with a window size of 32 data points, equivalent to 0.1 s since the transmitter is sending packets at a rate of 320 packets/second.

Once the outliers have been removed, we transform the acquired non-stationary signals into stationary signals by using the differencing procedure outlined in the "Initial Preprocessing" Section. To determine the stationarity of the signals, we selected some sample signals and tested them before and after the differencing process. Figure 10 shows a sample signal before and after applying the differencing procedure.

Figure 10 shows that the ADF test value before the test was equal to 0.388, which means that the signal is a non-stationary signal. After differencing the signal, the ADF test value is reduced to 0.001, which means a stationary signal is now available.

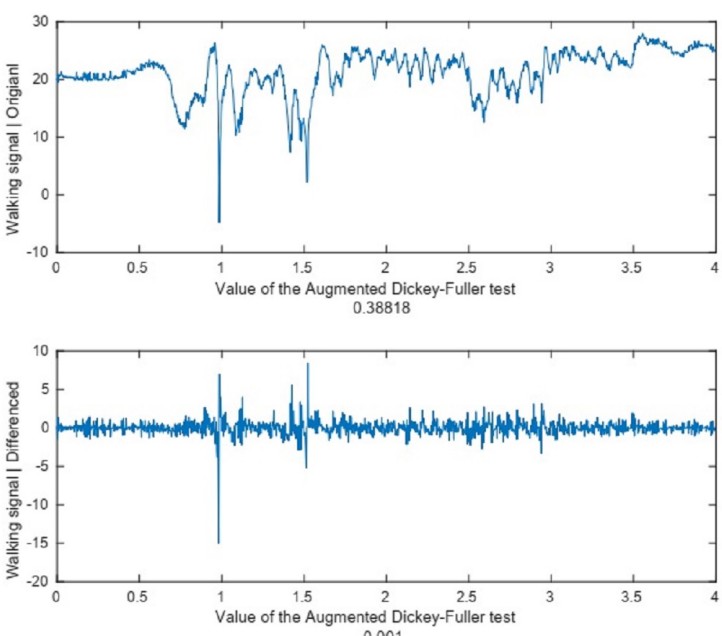

**Figure 10.** The effect of differencing on the ADF test.

Once a stationary signal has been acquired, we reduced the dimensionality of the available data by using the PCA algorithm. The PCA algorithm takes as input the gathered CSI sub-carriers and generates several PCA components. To determine the appropriate number of principal components to use, we relied on two metrics:

- The variance in the principal components. For this metric, we calculated the variance of the data in each of the principal components. Figure 11 shows the variance contained in each of the principal components generated from the first stream of the office environment. It shows no meaning in using all the principal components since there is no data variance in the components above the 10th component. We found out that 98% of the data variance is contained in the first eight principal components.

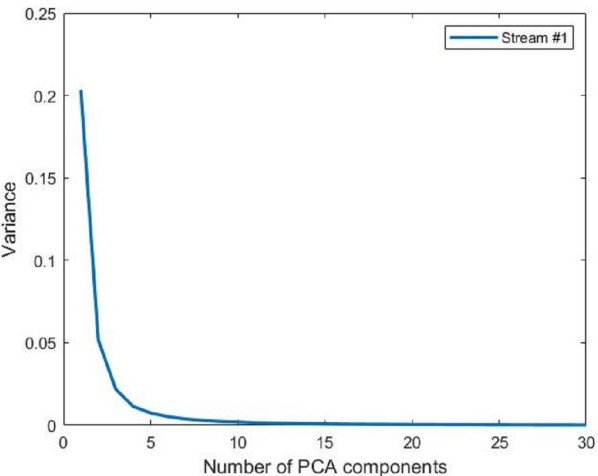

**Figure 11.** The variance in each PCA components.

- The error in activity segmentation. For this metric, we performed the segmentation procedure using various principal component combinations. We started with four components and stopped when we reached 15 components since most of the data are contained in the earliest components. The segmentation algorithm may lead to a segmentation error if it cannot determine the segment's starting or ending points. This

latter condition is used as a metric to determine how many principal components to use. Figure 12 shows the percentage of recorded activities that the segmentation algorithm was unable to extract a segment from. If we selected eight principal components, as the first evidence suggested, only 1% of the activities will not be segmented successfully. Such errors will be handled after the segmentation procedure as described in the "segmentation" Section.

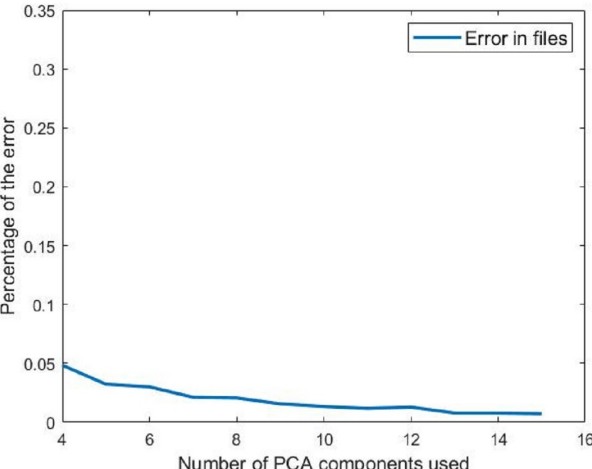

**Figure 12.** The percentage of activities that could not be segmented.

Based on the two metrics discussed before, we decided to use eight principal components as inputs to the segmentation process. They contain most of the information and produce reasonable segmentation errors compared to the work needed if more principal components were added.

The last data transformation we performed before the segmentation process was to smooth the signals. We used a Gaussian smoothing function with a window size of 48 points, which is equal to 0.15 s since the frequency by which packets arrive at the receiver is 320 packets/second. Figure 13 shows the effect of applying the smoothing function on the recorded signal.

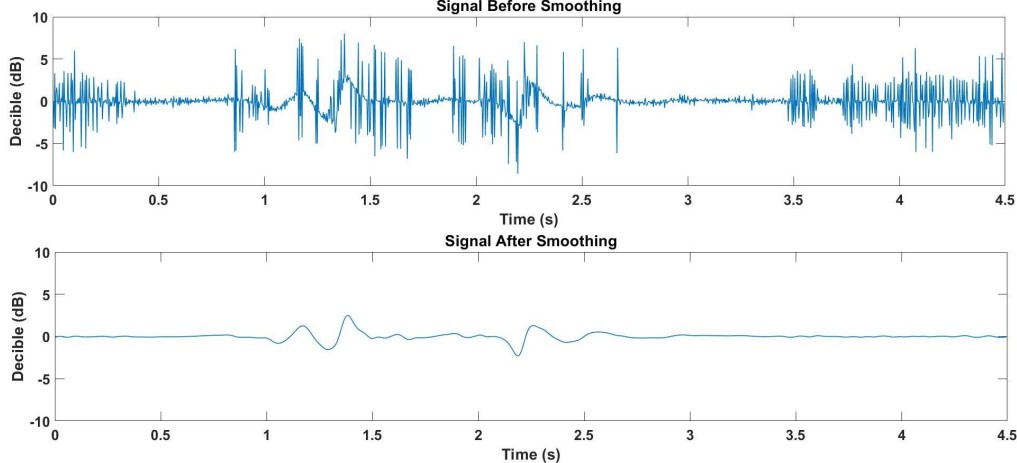

**Figure 13.** Effect of applying the smoothing technique on a recorded signal.

### 3.1.2. Segmentation

We applied the segmentation algorithm on the principal components generated from each of the three streams. This segmentation procedure identifies the starting and ending points for each of the activities performed by the subjects. Applying the segmentation algorithm can result in one of the following four cases:

I.    In the first case, the segmentation algorithm works correctly, and it was able to identify the start and the end of the activity within each of the communication streams. Thus, no further action is needed, and the starting and ending points are used to extract the segment;

II.    The second case happens when two streams are segmented successfully, while the third stream is not segmented. In this case, we take the average of the starting points from the two successful streams and the average of the ending points from the two successful streams and use them as the starting and ending points for the third unsuccessful stream;

III.    The third case occurs when only one stream is segmented successfully, while no segments were detected for the other two streams. In such a case, we assume that the segments for the two streams are the same as the segment for the stream we successfully found the segment in it; and

IV.    The last case comes about when the segmentation algorithm does not successfully extract the activity segment in any of the streams. The index that represents the beginning of the segment can then be computed as follows:

$$S = \frac{MIN(index(H), index(L))}{2}, \tag{6}$$

where $S$ is the index of the segment beginning, $H$ is the point associated with the highest value of the signal, and $L$ is the point associated with the lowest value of the signal.

On the other hand, the calculations for the segment endpoint are expressed as follows:

$$E = A - \frac{A - MAX(index(H), index(L))}{2}, \tag{7}$$

where $E$ is the index of the segment end and $A$ represents the index of the end of the signal. In other words, the segment end index is the midpoint between the end of the signal and the furthest occurrence of either the highest-peak or the lowest-peak values in that signal.

### 3.1.3. Post-Segmentation Preprocessing Stage

Using the points extracted from the segmentation process, the signal portion representing the activity is cropped and manipulated using the same processing procedure described in the "Before Segmentation Processing" section.

It is worth mentioning that removing the outliers, converting the signal to a stationary signal, and smoothing the signal can be straightforward. However, determining the best number of principal components is considered challenging. In the pre-segmentation processing stage, we determined the number of principal components by looking at the variance and at the number of signals that fall within the fourth case of the segmentation process that is described in the "segmentation" section. Using the percentage of segmentation failure is not appropriate since we are working with post-segmentation signals. To determine the number of principal components to use, we relied on the following two indicators: First, we calculated the variance available in each of the principal components in each communication stream. The variance results are shown in Figure 14. These results show that most of the data are contained in the first six components. In particular, the first six components contain 99.6% of the data as shown in Figure 14.

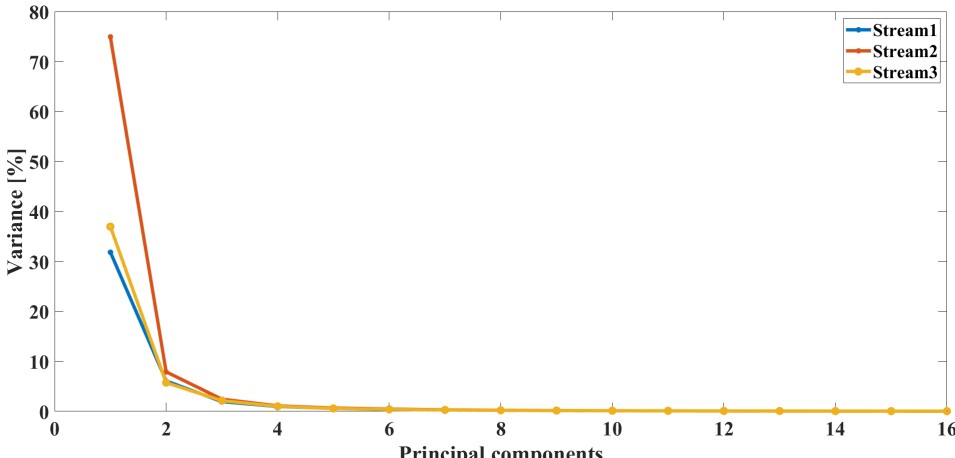

**Figure 14.** Variance in principal components post segmentation.

The second indicator we used to determine the number of components to use is by trying different principal components ranging from 4 to 9 components. We calculated the system's accuracy for each set of components when a different number of features is used. The results are shown in Figure 15. The results clearly show that using six principal components yields better results, in terms of accuracy, than using any other number of principal components. Based on the results shown in Figures 14 and 15, we are using six principal components in the experiments hereafter.

Figure 16 shows the different transformations that occur on the extracted segment. The signal plotted in Figure 16 is the signal's segment of a subject picking up a pen from the ground. Figure 16a shows the segment before applying any of the three-stage preprocessing procedure. The first stage will be removing the outliers using the Hampel filter as shown in Figure 16b. Once the outliers have been removed, the signal will be transformed into a stationary signal using the differencing procedure. The ADF test for this signal before differencing is equal to 0.5218, which was reduced to 0.001 after applying the differencing algorithm indicating that it is now stationary. Using the obtained stationary signal, several principal components will be extracted. Figure 16d shows the second principal component extracted from the signal. Finally, the Gaussian filter is applied to smooth the signal as shown in Figure 16e.

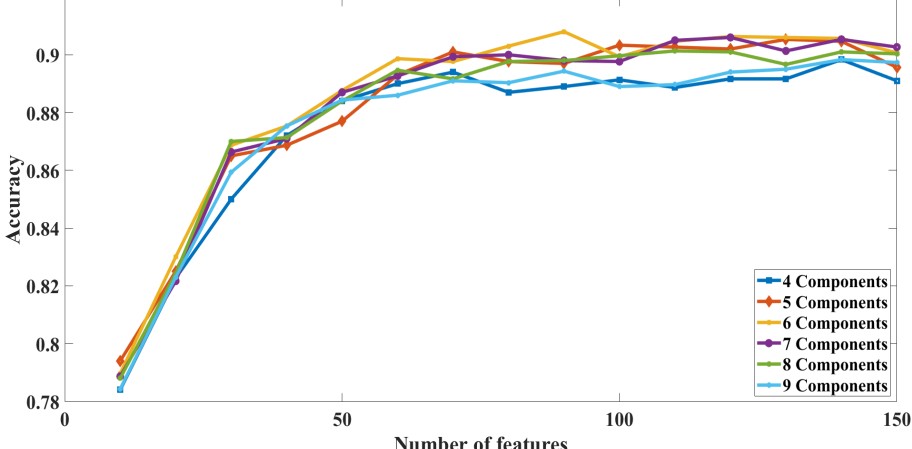

**Figure 15.** Effect of the number of PCA components used on the system's accuracy.

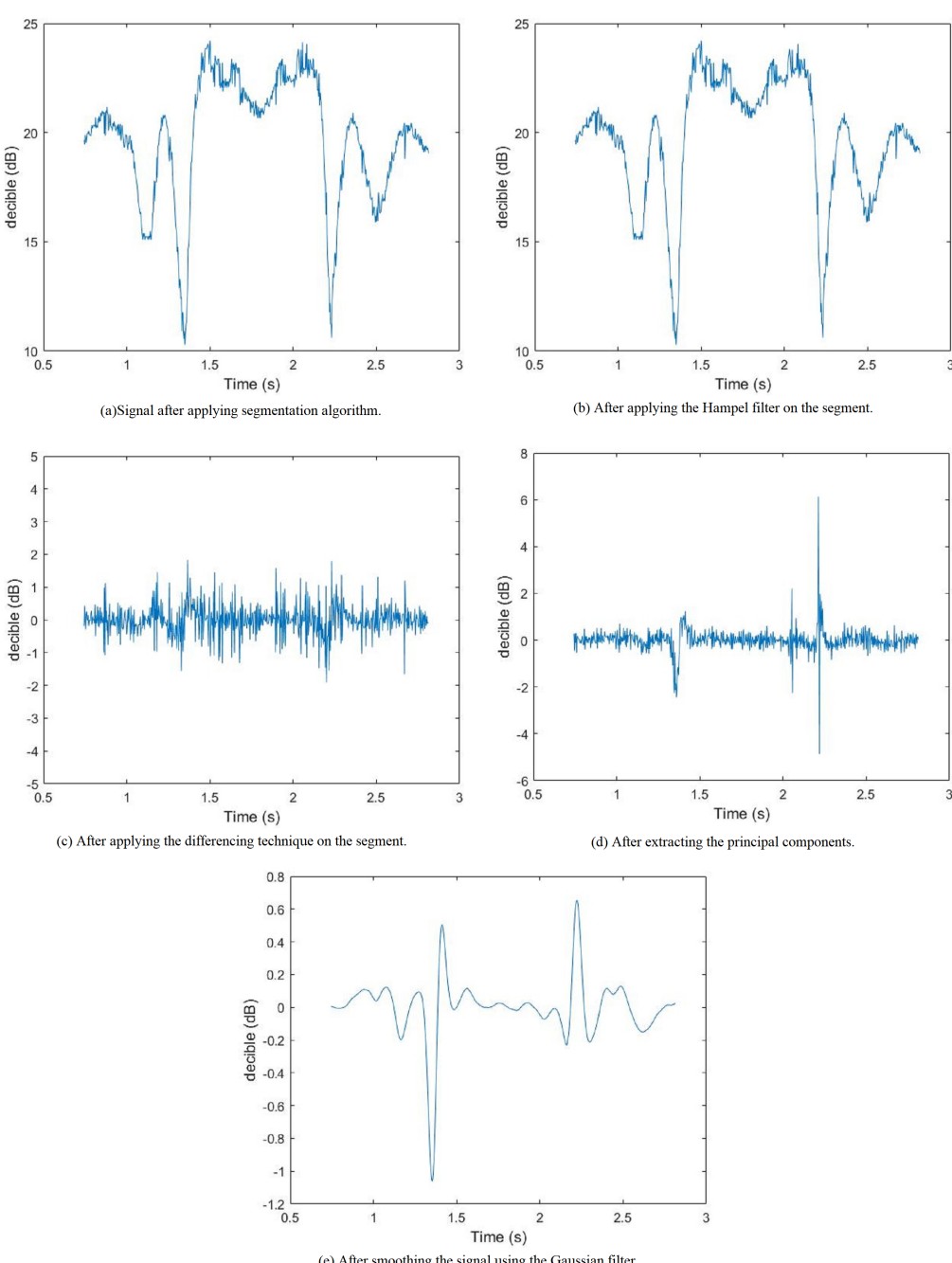

**Figure 16.** The output of each of the signal processing stages.

### 3.2. Feature Extraction and Selection

As discussed in the "Feature Selection" section, we are using the mRMR algorithm to determine the importance of the extracted features. To determine the number of features that yield the best activity recognition accuracy in general and fall detection in particular, we used six principal components as described in the "After segmentation processing" section. To perform the developed experiment, we trained our model multiple times by selecting a different set of features for each recorded signal. The features we selected are based on the importance score that was generated by the mRMR algorithm. The experiments' results are shown in Figure 17, which show that our proposed approach achieved its best performance when the number of used features is between 90 and 380 out of 565 features for both the hallway and the office environments. It is worth mentioning that tuning the parameters of the SVM classifier has the potential of further enhancing the performance of our proposed approach as described in the "Data Preparation" Section. For the office environment, the

feature selection results show that using 250 features out of the 565 features (i.e., 44.24% of the features) yields an accuracy of 91.23%. For the hallway environment, the feature selection results show that using 150 features out of the 565 features (i.e., 26.55% of the features) yields an accuracy of 86.4%. When the data from both environments are used to train the model, the results of the feature selection show that using 410 features (i.e., 72.56% of the features) achieved an accuracy of 88.8%.

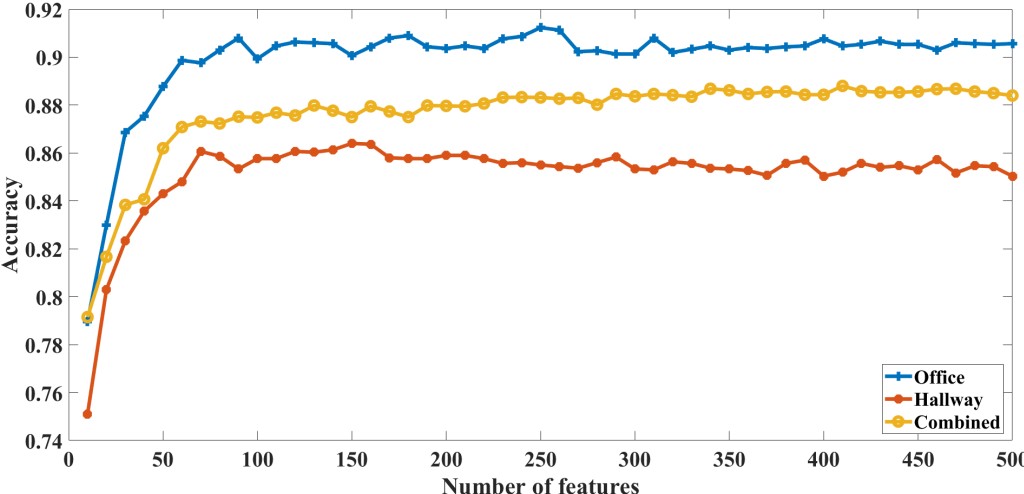

**Figure 17.** The effect of the number of features on the overall system's accuracy.

The aforementioned observations suggest the following: First, the number of features that yield the highest accuracy differs when the activities are performed in different environments. Second, the accuracy of the system depends on the performed activities as well as the noise in the environment. For example, when the participant moved from the office environment to the hallway environment, the accuracy of the system has decreased and the number of needed features has increased.

The selected features with the highest mRMR score that yielded the best accuracy for both the office and the hallway environments are summarized in Table 4. We also included the features with the highest mRMR scores when we combine the data from both the office and the hallway environments.

**Table 4.** Selected features.

| Set of Features | Number of Features Used for the Office Environment | Number of Features Used for the Hallway Environment | Number of Features Used When Both Environments Are Mixed |
|---|---|---|---|
| Time series features | 17 | 13 | 26 |
| SNR | 1 | 1 | 1 |
| Frequency features | 12 | 11 | 14 |
| Statistical Tests | 26 | 20 | 33 |
| Correlation features | 31 | 19 | 45 |
| Causation test | 2 | 1 | 2 |
| Residual Heteroscedasticity | 0 | 0 | 0 |
| Cointegration features | 3 | 2 | 3 |
| Instantaneous frequency | 17 | 9 | 23 |
| Spectral Entropy | 18 | 10 | 23 |
| Audio features | 123 | 64 | 240 |
| Total Number of features | 250 | 150 | 410 |

*3.3. Classification Results*

To validate the performance of our proposed approach, we have developed five evaluation scenarios based on the utilized dataset to train and test the SVM model. In particular,

the first evaluation scenario utilizes the dataset recorded in the office environment to train and test the SVM model, the second evaluation scenario utilizes the dataset recorded in the hallway environment to train and test the SVM model, the third evaluation scenario utilizes the datasets recorded in the office and hallway environments to train and test the SVM model, the fourth evaluation scenario utilizes the dataset recorded in the office environment to train the SVM model and the dataset recorded in the hallway to test the trained SVM model, and the fifth evaluation scenario utilizes the dataset recorded in the hallway environment to train the SVM model and the dataset recorded in the office environment to test the trained SVM model.

Moreover, for each of the first three evaluation scenarios, a leave-one-subject-out cross-validation (LOSO-CV) procedure is employed to train and test the constructed SVM classifiers. Specifically, the SVM classifier is trained using the feature vectors extracted from all subjects except one subject that is used to test the SVM classifier. This procedure is repeated $n$ times, where $n$ represents the number of subjects in the dataset used in each of the three evaluation scenarios, and the average performance is computed over the repetitions.

To quantify the results obtained for each activity, the metrics shown in Table 5 are used, where $\beta$ in the F1 score metric refers to the importance of the precision metric over the TPR metric. In our work, we fixed the value of $\beta$ to be equal to 1 to indicate that precision and TPR are equally important.

**Table 5.** The employed performance evaluation metrics. TP, TN, FP, and FN represent the true-positive, true-negative, false-positive, and false-negative values, respectively.

| Metric | Equation | |
|:---:|:---:|:---:|
| True-Positive Rate (TPR) | $\dfrac{TP}{TP + FN}$ | (8) |
| False-Positive Rate (FPR) | $\dfrac{FP}{TN + FP}$ | (9) |
| Precision | $\dfrac{TP}{TP + FP}$ | (10) |
| F1 Score | $(1 + \beta^2) \times \dfrac{Precision \times TPR}{(\beta^2 \times Precision) + TPR}$ | (11) |

### 3.3.1. Results of the First Evaluation Scenario

The dataset of the office environment was recorded from 10 individuals. Hence, the LOSO-CV procedure is applied by using the data of nine individuals to train the SVM classifier and the data of the remaining individual are used to test the SVM classifier. This process is repeated 10 times, such that each time the data for a different individual are used to test the SVM classifier and the data for the remaining individual are used to train the SVM classifier. The final results are obtained by averaging the results gathered from the 10 repetitions of the LOSO-CV process. The dataset size used for this experiment is equal to 3000 observations acquired from all 10 subjects. Each time we train the SVM classifier, we used 2700 observations, and the remaining 300 observations were left to test the developed SVM classifier.

The confusion matrix generated based on using the dataset of the office environment is shown in Table 6, which shows that a 91.27% overall activity recognition accuracy can be achieved. The statistics for each of the activities can be found in Table 7. The results show that, to detect a falling activity, a true-positive rate of 88% and a false-positive rate of 2.73% were achieved.

**Table 6.** The confusion matrix obtained for the first evaluation scenario.

|  |  | **Predicted Activity** | | | | | |
|---|---|---|---|---|---|---|---|
|  |  | A1 | A2 | A3 | A4 | A5 | A6 |
|  | A1 | 3456 | 24 | 3 | 33 | 78 | 6 |
|  | A2 | 18 | 1056 | 0 | 39 | 54 | 33 |
| **Performed Activity** | A3 | 0 | 6 | 1194 | 0 | 0 | 0 |
|  | A4 | 9 | 90 | 0 | 996 | 51 | 54 |
|  | A5 | 30 | 42 | 0 | 99 | 1020 | 9 |
|  | A6 | 9 | 39 | 0 | 54 | 6 | 492 |

**Table 7.** The results of our proposed approach obtained for the first evaluation scenario.

| Activity | A1 | A2 | A3 | A4 | A5 | A6 |
|---|---|---|---|---|---|---|
| True Positive Rate | 96.00% | 88.00% | 99.50% | 83.00% | 85.00% | 82.00% |
| False Positive Rate | 01.37% | 02.73% | 00.04% | 03.02% | 02.56% | 01.30% |
| Precision | 98.13% | 84. 01% | 99.75% | 81.57% | 84.37% | 82.83% |
| F1 Score | 97.05% | 85.96% | 99.62% | 82.28% | 84.68% | 82.41% |

### 3.3.2. Results of the Second Evaluation Scenario

Similar to the experiment performed in the office environment, we only used the dataset collected in the hallway environment to train and test the classification model. A LOSO-CV procedure was adopted to test the effectiveness of our model. The dataset size used for this experiment is equal to 3000 observations acquired from all 10 subjects. Each time we train the SVM classifier, we used 2700 observations, and the remaining 300 observations were left to test the developed SVM classifier. An overall accuracy of 86.53% was obtained, with the walking activity achieving the highest true-positive rate of 96.5% and picking up the pen from the ground achieving the lowest true-positive rate of 54%. By examining the results, one can observe a reduction in the system's performance compared to the experiments performed for the data collected in the office environment. This degradation in the performance can be traced back to the following reasons: First, the distance between the transmitter and the receiver in the hallway environment is longer than the distance between the transmitter and receiver in the office environment. This in turn can cause the received signals to be a little weaker due to fading effects. Secondly, we had no control over the environment since the experiment was performed in a public university hallway. The confusion matrix generated based on using the dataset of the hallway environment is shown in Table 8, and the detailed results are provided in Table 9. In this new environment, falling can be detected with a TPR of 87.75%.

**Table 8.** The confusion matrix obtained for the second evaluation scenario.

| | | Predicted Activity | | | | | |
|---|---|---|---|---|---|---|---|
| | | A1 | A2 | A3 | A4 | A5 | A6 |
| | A1 | 3411 | 27 | 9 | 42 | 84 | 27 |
| | A2 | 45 | 1053 | 0 | 30 | 69 | 3 |
| Performed Activity | A3 | 6 | 36 | 1158 | 0 | 0 | 0 |
| | A4 | 42 | 69 | 0 | 936 | 93 | 60 |
| | A5 | 102 | 102 | 0 | 63 | 906 | 27 |
| | A6 | 54 | 69 | 0 | 111 | 42 | 324 |

**Table 9.** The results of our proposed approach obtained for the second evaluation scenario.

| Activity | A1 | A2 | A3 | A4 | A5 | A6 |
|---|---|---|---|---|---|---|
| True Positive Rate | 94.75% | 87.75% | 96.50% | 78.00% | 75.50% | 54.00% |
| False Positive Rate | 05.38% | 04.31% | 00.14% | 03.47% | 04.02% | 01.54% |
| Precision | 93.20% | 77.65% | 99.23% | 79.19% | 75.88% | 73.47% |
| F1 Score | 93.97% | 82.39% | 97.85% | 78.59% | 75.69% | 62.25% |

### 3.3.3. Results of the Third Evaluation Scenario

As with the experiments performed when the data collected in the office or the hallway environment, in this experiment, we randomly selected a single subject to test the model's performance that was built using the data from the remaining subjects. Since we are combining the data from the office and the hallway environments, we have 20 subjects in this experiment. 20-fold cross-validation was performed with a different subject selected in each of the folds to test the model built using the data from the remaining 19 subjects. The dataset size used for this experiment is equal to 6000 observations acquired from all 20 subjects. Each time we train the SVM classifier, we used 5400 observations, and the remaining 600 observations were left to test the developed SVM classifier. The experiments showed an average accuracy for activity recognition of 88.82%, with falling detection reaching 87.25% TPR. The confusion matrix is shown in Table 10 and the calculated performance metrics are provided in Table 11.

**Table 10.** The confusion matrix obtained for the third evaluation scenario.

| | | Predicted Activity | | | | | |
|---|---|---|---|---|---|---|---|
| | | A1 | A2 | A3 | A4 | A5 | A6 |
| | A1 | 6951 | 57 | 6 | 54 | 108 | 24 |
| | A2 | 81 | 2094 | 0 | 87 | 120 | 18 |
| Performed Activity | A3 | 3 | 39 | 2358 | 0 | 0 | 0 |
| | A4 | 84 | 150 | 0 | 1974 | 123 | 69 |
| | A5 | 153 | 183 | 0 | 135 | 1908 | 21 |
| | A6 | 78 | 144 | 0 | 216 | 60 | 702 |

**Table 11.** The results of our proposed approach obtained for the third evaluation scenario.

| Activity | A1 | A2 | A3 | A4 | A5 | A6 |
|---|---|---|---|---|---|---|
| True Positive Rate | 96.54% | 87.25% | 98.25% | 82.25% | 79.50% | 58.50% |
| False Positive Rate | 04.23% | 03.96% | 00.04% | 03.39% | 02.84% | 00.86% |
| Precision | 94.57% | 78.52% | 99.75% | 80.05% | 82.28% | 84.17% |
| F1 Score | 95.55% | 82.65% | 98.99% | 81.13% | 80.86% | 69.03% |

### 3.3.4. Results of the Fourth Evaluation Scenario

In the experiment, we used the data collected in the office environment to build the classification model and the data collected in the hallway environment to test the model. The dataset collected for this experiment is comprised of 6000 observations. In addition, 3000 observations from the office environment were used to train the model, and 3000 observations from the hallway environment were used to test the model. In this experiment, an average activity recognition accuracy of 79.93% was achieved. Regarding fall detection, the TPR was reduced to 77.00% compared to 90.00% when data from the two environments are used with the data from the same subject being integrated within the training dataset. The confusion matrix is shown in Table 12 while the performance metrics are shown in Table 13.

**Table 12.** The confusion matrix obtained for the fourth evaluation scenario.

| | | Predicted Activity | | | | | |
|---|---|---|---|---|---|---|---|
| | | A1 | A2 | A3 | A4 | A5 | A6 |
| | A1 | 3501 | 30 | 12 | 0 | 39 | 18 |
| | A2 | 96 | 924 | 0 | 9 | 153 | 18 |
| **Performed Activity** | A3 | 3 | 111 | 1077 | 3 | 0 | 6 |
| | A4 | 165 | 183 | 3 | 516 | 183 | 150 |
| | A5 | 159 | 51 | 0 | 18 | 951 | 21 |
| | A6 | 72 | 75 | 0 | 84 | 144 | 225 |

**Table 13.** The results of our proposed approach obtained for the fourth evaluation scenario.

| Activity | A1 | A2 | A3 | A4 | A5 | A6 |
|---|---|---|---|---|---|---|
| True Positive Rate | 97.25% | 77.00% | 89.75% | 43.00% | 79.25% | 37.50% |
| False Positive Rate | 11.82% | 06.70% | 00.24% | 01.68% | 07.68% | 02.97% |
| Precision | 87.61% | 67.25% | 98.63% | 81.90% | 64.69% | 51.37% |
| F1 Score | 92.18% | 71.79% | 93.98% | 56.39% | 71.24% | 43.35% |

### 3.3.5. Results of the Fifth Evaluation Scenario

In the experiment, we used the data collected in the hallway environment to build the classification model and the data collected in the office environment to test the model. The dataset collected for this experiment is comprised of 6000 observations. In addition, 3000 observations from the hallway environment were used to train the model, and 3000 observations from the office environment were used to test the model. In this experiment, an accuracy of 87.23% was achieved, with the falling activity reaching a 74.50% TPR. The confusion matrix is shown in Table 14 while the performance metrics are shown in Table 15.

**Table 14.** The confusion matrix obtained for the fifth evaluation scenario.

| | | Predicted Activity | | | | | |
|---|---|---|---|---|---|---|---|
| | | A1 | A2 | A3 | A4 | A5 | A6 |
| | A1 | 3375 | 45 | 6 | 54 | 99 | 21 |
| | A2 | 36 | 894 | 15 | 90 | 90 | 75 |
| **Performed Activity** | A3 | 0 | 0 | 1200 | 0 | 0 | 0 |
| | A4 | 0 | 75 | 3 | 1020 | 48 | 54 |
| | A5 | 6 | 84 | 0 | 96 | 990 | 24 |
| | A6 | 3 | 63 | 0 | 159 | 3 | 372 |

**Table 15.** The results of our proposed approach obtained for the fifth evaluation scenario.

| Activity | A1 | A2 | A3 | A4 | A5 | A6 |
|---|---|---|---|---|---|---|
| True Positive Rate | 93.75% | 74.50% | 100.00% | 85.00% | 82.50% | 62.00% |
| False Positive Rate | 01.00% | 03.70% | 00.36% | 05.52% | 03.38% | 02.27% |
| Precision | 98.68% | 77.00% | 98.04% | 71.88% | 80.49% | 68.13% |
| F1 Score | 96.15% | 75.73% | 99.01% | 77.89% | 81.48% | 64.92% |

## 4. Discussion

The results presented in the "Results" Section signify the following observations: First, the environment in which the activities were performed affects the achieved recognition accuracy. This is illustrated by the difference in the accuracies obtained for the office and hallway environments. More specifically, in the office environment, the distance between the transmitter and the receiver was 3.6 m, while the distance between the transmitter and receiver in the hallway environment was 7.6 m. This difference in the distance between the transmitter and receiver in the two environments resulted in a decrease of 4.74% in the accuracy achieved using the data recorded in the hallway environment. Second, more data do not necessarily mean better performance. Specifically, the results indicate that using only six principal components instead of using all the principal components achieves better performance, which is shown in Figure 15. In addition, using all the features does not mean better performance. Particularly, Figure 17 shows that using a subset of selected features achieves better performance than using all the features.

Third, the confusion matrices clearly show that for both environments the walking activity achieved the highest accuracy with 99.5% TPR in the office environment and 96.5% TPR in the hallway environment. This can be attributed to the fact that the pattern of walking is highly different from the pattern of falling, sitting on a chair, or standing still. Specifically, when the subjects perform the walking activity, they cover the entire distance between the transmitter and the receiver. On the other hand, when the subjects perform the falling, sitting on a chair, and picking up a pen from the ground, they remain in the same place, limiting their movement pattern. Thus, activities such as falling, picking up a pen from the ground, and sitting on a chair have more incorrect predictions due to the relatively high similarity among these activities, while the walking activity has less incorrect predictions, which is translated to higher recognition accuracy due to their distinct movement pattern.

### 4.1. Performance Comparison

Table 16 provides a comparison between the work presented in this paper and other similar papers. Table 16 shows that our proposed approach was able to achieve higher recognition accuracy compared to other existing approaches, which suggests the feasibility of the proposed approach to recognize human activities using CSI signals. We contribute the good performance of our work to the following reasons. First, in this work, we developed a four-staged procedure that achieves the following: removing outliers, converting the signal from non-stationary to stationary, reducing the size of the available data, and smoothing the signals. Second, the four-stage signal denoising procedure was performed twice: before and after segmentation. In addition, finally, a coarse and fine grid search was performed to determine the optimal SVM parameters. Moreover, to the best of our knowledge, our study is one of the first studies that investigates the problem of HAR and fall detection using the LOSO-CV procedure, which is considered more challenging than traditional cross-validation procedures.

**Table 16.** Comparison with other existing approaches.

| Method | Number of Subjects | Number of Environments | Number of Activities | Data Type | Classifier | Overall Activity Detection Accuracy | Fall Detection Accuracy |
|---|---|---|---|---|---|---|---|
| [66] | Not Specified | 2 | 10 | CSI | CNN | Not specified | 90% |
| [67] | 10 | 3 | 6 | CSI | RNN | 90% | 93% |
| [63] | 3 | 3 | 10 | CSI | SVM | Not specified | 93% |
| [64] | 8 | 3 | 4 | CSI | SVM | Not specified | 90% |
| [65] | 7 | 1 | 11 | CSI | CNN-SVM | 90.90% | Not specified |
| [68] | 10 | 2 | 6 | CSI | CNN | 91.2% | Not specified |
| proposed approach | 20 | 2 | 6 | CSI | SVM | 91.27% | 96.16% |

*4.2. Limitations and Future Work*

The main focus of this study is to investigate the possibility of recognizing human activities in multiple environments within the same floor. The experimental results show that our proposed approach was able to accurately recognize the different performed human activities. Hence, the scenario that involves human activities performed in a multi-room/multi-floor environment was not investigated in the current study.

In the future, we intend to extend the work presented in this paper by experimenting with a unified set of features, which can be used to build a single classification model that can accurately identify the performed activities regardless of the environment. A more complex multi-room/multi-floor environment will be used as a data collection environment. Furthermore, different feature selection methods will be investigated to determine which of them provides the best performance accuracy.

**5. Conclusions**

Determining what activities are being performed without hindering the movements of the subjects is now much easier due to the utilization of the overflowing Wi-Fi signals in almost all places. This work proposed an approach to determine the activity being performed, including fall and non-fall activities, by extracting and analyzing the information contained within the CSI values of the OFDM subcarriers associated with the exchanged Wi-Fi signals. All the experiments in this work were applied to real data that are acquired from real subjects. We also present a segmentation algorithm to determine the portion of the signal which contains the performed activity. By processing the acquired signals, before and after the segmentation process, we achieved high activity recognition rates. Furthermore, we have compared the performance of the proposed method with the performance of well-known state-of-the-art activity recognition systems. The experimental results show that a 91% overall activity recognition can be achieved using the proposed method which includes denoising and segmentation processes.

**Author Contributions:** Conceptualization, B.A.A. and R.A.; methodology, B.A.A. and R.A.; software, M.M.A.; validation, B.A.A. and M.M.A.; formal analysis, B.A.A. and M.M.A.; investigation, B.A.A. and M.M.A.; resources, B.A.A. and M.M.A.; data curation, M.M.A.; writing—original draft preparation, B.A.A.; writing—review and editing, B.A.A., R.A., S.A. and M.I.D.; supervision, B.A.A.; project administration, B.A.A.; funding acquisition, B.A.A. and R.A.; Visualization, B.A.A. All authors have read and agreed to the published version of the manuscript.

**Funding:** This research was funded by Jordan University of Science and Technology Grant No. 20180032.

**Institutional Review Board Statement:** The study was conducted according to the guidelines of the Declaration of Helsinki, and approved by the Institutional Review Board of King Abdullah University Hospital (KAUH) and the Jordan University of Science and Technology (JUST) (19/110/2017).

**Informed Consent Statement:** Informed consent was obtained from all subjects involved in the study.

**Data Availability Statement:** The data presented in this study are openly available in [Mendeley] at [10.17632/v38wjmz6f6.2].

**Conflicts of Interest:** The authors declare no conflict of interest.

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
