# Peer review of "A CSI-Based Multi-Environment Human Activity Recognition Framework"

_applsci, doi:10.3390/app12020930_

Round 1

Reviewer 1 Report

The authors have presented a novel idea of classifying the human activities using the pattern changes in the wifi signals with the help of machine learning algorithms. I have following concerns for this paper;

Major Concern: Explain the relation / difference of your work with the following earlier work;

Lagashkin, Ruslan. "Human localization and activity classification by machine learning on Wi-Fi channel state information." (2020).

Minor concerns:

Improve Text visibility of Fig 2 and 3, 5,6

Improve Text visibility in Table 3 and 16

Improve Fig 14,15,16,17 text visibility and graph lines size

In Table 12 caption correct confusing matrix to confusion matrix

Author Response

We would like to thank you for your time and constructive comments regarding our paper “A CSI-based Multi-Environment Human Activity Recognition framework”. For your reference, the attached document provides point-to-point responses that explain how we addressed each of the raised comments. 

Reviewer 2 Report

The paper is very interesting and the experiment well presented with appropriate results discussions.
There are several minor writing (line 22) and grammar error (lines 70, 412, 442).
In the paragraph 54-78 there is a little incoherence in presenting the activities to be recognized.
Also, I would recommend bigger fonts for figures 2 and 3.
I think it would be easier to read and understand the presentation if the figures would be closer to their reference in text (see figure 15 and 16).

Author Response

(The authors gave the same response as above.)
